# DATA-DRIVEN CREATIVITY: AMPLIFYING IMAGINATION IN LLM WRITING

## ABSTRACT

During the alignment training of Large Language Models (LLMs), Reinforcement Learning from Human Feedback (RLHF) has proven to be effective in enhancing the model's alignment with human preferences. The RLHF approach requires human annotators to provide data representative of human preferences, aiding the model in advancing towards human-preferred outcomes. In this process, high-quality human preference data is both crucial and challenging to obtain. While many tasks, such as coding and mathematics, can now be more efficiently annotated through Artificial Intelligence Feedback (AIF), numerous tasks still necessitate human input to provide human preference signals. Particularly creative tasks are typical tasks that involving complex human preference. Here, we focus on creative writing tasks and investigate how to collaborate with annotators to acquire high-quality, superior data. We propose an expert-assisted data generation process, named Expert-Objective-Personal-Subjective (EOPS), that can efficiently obtain high-quality ordinal data with minimal human resources. We conduct experiments on three kinds of tasks, and experimental results validat the effectiveness of our method.

## 1 INTRODUCTION

In the alignment training process of LLMs, RLHF can effectively enhance the model's performance in aligning with human preferences (Ouyang et al., 2022; Bai et al., 2022). Numerous studies have validated the effectiveness of algorithms such as PPO and DPO based on preference data. However, due to the tedious and expensive nature of manual annotation, most works have attempted to acquire data more efficiently through AIF methods (Lee et al., 2023). These methods are typically applicable to relatively objective tasks in science and mathematics, such as logic reasoning and coding. For more complex generative tasks, such as creative writing (Wang et al., 2024), where there are no definitive answers and the creative generation requirements are difficult to evaluate objectively, AI-based annotations often fall short of the desired outcomes.

Our work focuses on how to collaborate with annotators to obtain high-quality data in the domain of open-ended generative tasks, with innovative writing as our chosen task. The first challenge in open-ended tasks is the absence of clear standard answers, which not only makes it difficult to use AI-assisted annotations but also poses challenges for annotators. Different populations may have varying preferences for open tasks, and excessive subjective preferences can introduce significant noise, making learning for the task difficult. To address this issue, we propose a decomposition scheme of Expert-Objective-Personal-Subjective (EOPS), incorporating a domain expert to provide professional opinions and standards in relatively objective areas. By quantifying these relatively objective standards, we can assist annotators in aligning with expert criteria in this part. The remaining portion that requires the introduction of subjective preferences is left to the annotators, allowing them to rank based on their own opinions. This decomposition approach aims to minimize noise while preserving the diversity of preferences.

Specifically, our data production process involves the generation of prompts, responses, and the acquisition of annotation rankings. This process requires the collaborative participation of design researchers, domain experts, and annotators. In the prompt generation phase, domain experts provide task decomposition, and researchers introduce AI-assisted methods to obtain a large amount of data. During the annotation process, domain experts offer quantifiable objective standards, while

annotators combine these standards with their understanding and preferences to provide the final ranking. We have implemented the entire process for producing partial-order data for creative writing.

We have validated the effectiveness of our method on the tasks of creative writing. On the Baichuan series of models, the data we produced significantly enhanced the models' capabilities in creative writing. We conducted tests on both in-domain and out-of-domain data, confirming the effectiveness of our approach. We also test our data on Qwen2 models (Yang et al., 2024a) and find that the data transfer in RLHF might not be effective.

In summery, our main contribution is to propose a whole pipline to annotate preference data for creative writing. Our EOPS pipline leverage the expert knowledge to reduce human noise. Our experimental results validate the effectiveness of our method.

Below, we list some related works in Sec. 2. Our method is shown in Sec. 3 and the experimental results are shown in Sec. 4.

## 2 RELATED WORK

RLHF (Ouyang et al., 2022; Bai et al., 2022; Zheng et al., 2023; Touvron et al.) method has been proved to be an effective method for LLM alignment. However, the collection of high quality data is the most important and hardest problem for using RLHF. Ouyang et al. (2022) and Bai et al. (2022) mentioned of using human annotators to collect preference data, but not much details are given.

Considering the efficiency and cost, many work turns to collect data using LLMs themselves. Lee et al. (2023) propose a method to use AI feedback to replace human feedback. Cui et al. (2023) uses responses from various models and employ GPT-4 to give scores. On specific tasks, such as math (Wang et al., 2023; Yang et al., 2024b) or reasoning (Havrilla et al.), AIF proves to gain some improvements. Burns et al. (2023) further explores the method when human knowledge is not enough for annotation.

However, tasks where human annotations are needed are much less explored. Creative writing can be considered as a typical example where LLM might not provide accurate signals for annotation. (Wang et al., 2024) conduct much work on creative writing. They construct preference pairs from previous constructed data, which is not a standard preference data generation process. Our work takes inspiration from this work, but we mainly focus on improving the performance through RLHF training. Thus we concentrate on preference data generation directly.

Besides RLHF, there are many other methods for alignment, such as supervised fine-tuning (Sanh et al., 2021) and Direct Policy Optimization (Rafailov et al., 2023). Different method might prefer different kinds of data. Our work mainly focus on RLHF data generation.

## 3 METHOD

Here we introduce the entire data production process proposed by us. This process integrates experts, annotators, and automation algorithms to enhance efficiency while ensuring that the data's partial order annotations are of high quality and distinctiveness. Our process consists of the following stages: task decomposition, prompt generation, response production, annotation rule generation, and the final data annotation process. We will elaborate on each part below.

### 3.1 TASK SEPERATION

The scope of creative tasks is vast, and human experts decompose creative writing tasks based on human experience. Since our annotation process involves annotators referencing annotation criteria, the way we decompose tasks here primarily focuses on experts breaking down the large task of creative writing into generation tasks, expansion tasks, and rewriting tasks based on the key points to be considered during annotation.

## 3.2 PROMPT GENERATION

In the production process of prompts, our principle is to make the prompts as rich and diverse as possible, and capable of mass production. Therefore, we consider using AI-assisted prompt generation methods. Firstly, we introduce human experts to further decompose each task, and then based on the various dimensions of the decomposition, use LLM to generate prompts.

Firstly, we divide the types of instructions contained in the prompts: (1) Context: This refers to the text related to the generated content that may be included in the prompt, such as the theme in story creation, or the original text in rewriting and expansion tasks; (2) Requirements related to the new generated text, or instructions, such as genre and style. Then, we enumerate the specific instructions that may be involved in these two types of instructions, obtaining a variety of specific instructions related to each task. For each type of instruction, we further sample dozens of elements. We list examples of the types of instructions and elements involved in three tasks in the Table **??**.

| Task | Instruction Type | Specific instruction | Examples |
|------|------------------|----------------------|----------|
| Short story generation | context | topic | Apple |
| Short story generation | requirements | story type | Detective story |
| Short story generation | requirements | language style | Realism |
| Expand writing | context | topic | Love |
| Expand writing | context | background | Ancient dynasty |
| Expand writing | context | character | Soldier |
| Expand writing | requirements | style | Romanticism |
| Expand writing | requirements | details | Environmental description |
| Style transfer | context | topic | Friendship |
| Style transfer | context | background | Magic world |
| Style transfer | context | region | Europe |
| Style transfer | requirements | target style | Classical Chinese |
| Style transfer | requirements | author style | Li Bai |

Table 1: Three types of creative writing tasks and examples of elements for prompt generation.

Ultimately, in the process of generating prompts, we first sample the instructions, and for the sampled instructions, we further sample one or more elements. By concatenating these instructions and elements, we enable the LLM to generate a smooth and natural prompt. Most instructions follow the effect that qualified LLMs can achieve. We mainly conducted prompt sampling on the Baichuan series of models.

## 3.3 RESPONSE GENERATION

RLHF is a learning approach aligned with human preferences, where acquiring human preferences is the most important and challenging part. If annotators are all domain experts, we can directly use expert preferences. However, due to conditional limitations, we may often need non-professionals to participate in annotation, making a scheme where experts design annotation rules to assist annotators more feasible.

We propose an Expert-Objective-Personal-Subjective (EOPS) method, which reduces annotation noise through expert quantification of objective criteria and a labeling scheme where annotators combine objective criteria for subjective ranking. EOPS can maximize the quality of data annotation under limited annotation resources.

EOPS requires experts to extract relatively objective indicators for innovative writing tasks and provide scoring evaluation criteria. The design of the indicators is crucial. Incomplete rules may lead annotators to label through shortcuts. For example, if the labeling standard only emphasizes the constraint of word count, annotators may only focus on the word count constraint, ignoring the more critical content evaluation. Based on such shortcut methods, the partial order data trained by RM may achieve high accuracy on its own same-distribution validation set, but it is difficult to obtain significant improvements in writing quality.

For innovative writing tasks, we break down the objective indicators into: "Instruction Theme Extraction", "Correctness Score", and "Language Score". We introduce the specific norms for each item below. Annotators need to make clear scoring judgments based on the expert opinions on these items. On the basis of objective indicator scoring, we hope that annotators will compare all responses to the same prompt overall, introduce their own subjective preferences during the comparison process, and finally give the "Response Order". This ranking will ultimately become the basis for constructing our partial order data. Additionally, to avoid annotators skipping subjective comparisons and sorting through shortcuts, as well as to facilitate data verification, we need annotators to add "comments" to each response to further explain the sorting reasons.

## 3.4 ANNOTATION STANDARD

Reinforcement Learning from Human Feedback (RLHF) is a learning approach aligned with human preferences, where obtaining human preferences is both the most important and challenging part. If annotators are all domain experts, we can directly use expert preferences. However, due to resource constraints, we may often need non-experts to participate in annotations. Therefore, a more feasible solution is to have experts design annotation rules to assist annotators in the annotation process.

### 3.4.1 INTENTIONS CONVEYING

In the field of artificial intelligence, a prompt is a text or instruction used to guide a model in generating specific outputs. It allows users to interact with the model by providing specific contexts, thereby obtaining more accurate and targeted results. A good prompt should possess clarity, relevance, and accuracy, effectively conveying the user's intentions and guiding the model to generate content that meets expectations.

The prompt is completed by the model under specific scenarios and instructions. To better verify whether the generated response is of high quality and completeness, annotators need to extract key elements from the instructions, namely extracting keywords. The thematic elements of the instruction generally have no less than three, with partial overlap before and after, and the key points are taken, which is to assist subsequent verification, adhering to the principle of "more is better, less is not enough."

For example, w consider below prompt:

> Please write a campus novel about a student who is strongly influenced by their family background in school and how they interact with different classmates and clubs. Describe how the student deals with the complex emotions of adolescence and the difficult topic of sex education. Try to incorporate elements of the blacksmith club, such as scenes where the student council discusses forging techniques, metal material selection, or the manufacturing process of iron products. This will be a rich story that blends campus life, family education, and the inheritance of craftsmanship.

The extracted themes should be: "Dealing with the complex emotions of adolescence and sex education", "Elements of the blacksmith club", "Campus life, family education, and the inheritance of craftsmanship".

### 3.4.2 CORRECTNESS

The correctness score, or "answer relevance", involves extracting keywords from the instruction theme and then finding and marking corresponding answers within the response to check the degree of correspondence and obtain the corresponding correctness score. This helps us test the extent to which the response follows the instruction and assign corresponding scores.

We give an example of generated response (Corresponding to prompt above).

"Steel Flowers in Full Bloom Spring"

Chapter 1: New Beginnings and Challenges On a sunny September morning, sunlight filtered through the lush leaves of towering trees, casting shadows across the campus of a renowned high school. This was a place where dreams took root, and every young heart carried with it hopes for a bright future, Lin Ran among them. He carried an old brown backpack, and beneath his jet-black hair, his deep eyes seemed to hold countless stories.

Lin Ran's family background was rather unique—his father was a skilled blacksmith who ran a small workshop with a reputable name in the local area. Growing up in such an environment, Lin Ran developed a special affection for steel, not only mastering basic blacksmithing skills but also understanding the persistence and love behind the craftsmanship. However, this family background made him somewhat out of place among his classmates, who were more concerned with academic achievements and social trends than traditional handcraft arts.

Despite this, Lin Ran remained undaunted. He believed that everyone's birth had its unique meaning and mission, so he decided to face the challenges brought by high school life in his own way. He joined the school's only blacksmithing club and became one of its most active members. Whenever the topic of handcraft arts came up in class, Lin Ran always offered insightful opinions and suggestions, and he could even demonstrate his blacksmithing skills on the spot, earning the respect of many classmates.

Chapter 2: Sparks of Youth As the semester progressed, Lin Ran began to interact with more peers of his age. He discovered that everyone had their own troubles and joys, secrets, and aspirations for the future. In an accidental conversation, Lin Ran learned that his roommate Li Hao was struggling with sex education. Adolescents often shy away from such topics, but Lin Ran used a mature mindset to understand and help Li Hao. Together, they discussed the importance of sex education and how to correctly face and handle these complex emotions. It was this honest exchange that led to the two becoming inseparable friends.

At the same time, the school's blacksmithing club was bustling with various activities. The annual "Steel Tempered into Art" competition was approaching, which served as an important stage for showcasing the skills and creativity of club members. Lin Ran and Li Hao teamed up to prepare a creative ironwork piece. Through the process of creation, they not only strengthened their friendship but also gradually found their respective positions and values in artistic creation.

Chapter 3: Blossoming Dreams Time flew by, and the day of the "Steel Tempered into Art" competition finally arrived, with the entire campus filled with tension and excitement. The diversity of the entries was breathtaking, ranging from exquisite small household items to grand sculptures, each piece embodying the sweat and wisdom of its creator. Among these exquisite artworks, the iron flowers created jointly by Lin Ran and Li Hao stood out, resembling steel flowers in full bloom, dazzling and vibrant, yet exuding a sense of indomitable strength. When asked about their creative inspiration, Lin Ran honestly stated that it originated from his father's teachings about the malleability and endurance of steel, symbolizing the colorful brilliance of youth and dreams. These words not only deeply touched every audience member but also won their work the top prize! At this moment, all their efforts were rewarded.

Annotators need to check the presentation of the themes extracted from prompt in the response. For example, for the theme "Dealing with complex emotions and sex education during adolescence", we can find "Lin Ran learned that his roommate Li Hao was struggling with sex education. Adolescents often shy away from such topics, but Lin Ran used a mature mindset to understand and help Li Hao. Together, they discussed the importance of sex education and how to correctly face and handle these complex emotions." Thus, we can count the number of themes that are followed.

We use a 5-point full-score decrement system, deducting one point for each missing theme. In the response corresponding to the instruction, all the aforementioned themes are presented, so we choose "5 points" — a complete match, selecting 5 points.

### 3.4.3 LANGUAGE SCORE

This language score is relatively dependent on the annotator's sense of language judgment of this text. Therefore, we refine such an abstract standard into various dimensions such as language na-

ture, literariness, and plot richness to help give a comprehensive language score and let us judge the language expression ability generated by machines. The language score is also a 5-point full-score. There are 7 elements are included for scoring: (1) richness of plot, (2) characterization, (3) expression, (4) depth of the theme, (5) creative imagination, (6) emotional resonance, and (7) logical rigor. The annotator should take all these into consideration and choose a score from 1-5.

### 3.4.4 RANKING

Each instruction has X answers, and rankings from 1 to X need to be provided. Based on the above "correctness score" and "language score", annotators need to give a final ranking based on their own preferences. Their own preference includes the annotator's literary understanding and thinking.

For creative writing, especially for the extensive and profound Chinese characters, there are many dimensions for evaluation. We hope to obtain evaluations in other aspects besides "correctness score" and "language score". In this way, we can further judge the generation quality of the machine, and also enable human experts and algorithm engineers to have a more detailed understanding of the quality of this batch of data and make more optimizations. Here, we select some examples of "remarks".

Here is an example of remark:

> Eyes finally stay on that particular window - "it always opens by itself at midnight", "begins to suspect that there may be an unknown secret room hidden in the villa, and that secret room may be the key to solving the whole mystery" and other expressions lack logic, the story line is less coherent, there is a logical vulnerability, "It is the mysterious window depicted above and the scene outside the window" that the murals hide secrets, and "The mysterious window and the whispers outside the window are just a deception set up by the missing person to cover his whereabouts". Paradoxically, the story does not mention finding the missing person, which does not match the requirements of the original text, and the disclosure of the turning point is less reflected and the story is weaker.

### 3.5 DATA ANNOTATION PROCESS

In the annotation process, 5 to 7 annotators participate. During the annotation process, experts, algorithms, and annotators will communicate closely to solve the difficulties encountered in annotation. We will collect the annotation suggestions put forward by annotators during the annotation process and further integrate them into the annotation rules to improve the entire annotation process.

## 4 EXPERIMENTS

With preference generated above, we follow RLHF training method to validate the final performance improvement. Our experiments mainly mainly conduct in two parts. In the first part, we mainly run RLHF on the short story generation task. For the second part, we extend tasks to the expand writing and style transfer tasks to validate the effectness of our data generation pipline.

### 4.1 SHORT STORY GENERATION EXPERIMENTS

### 4.1.1 SETTING

Concentrating on the story generation task, we generate 594 prompts and get 4157 responses. After our EOPS process, we construct pairs with the final ranking and get 12435 preference pairs.

For this experiment, we use an old version of Baichuan 4 model, which we called BC-old. We use these data to fine-tune one relatively small Baichuan model to get a RM model. Notice that we combine with other types of preference data to ensure the general ability during training. We then use this RM to conduct PPO training from BC-old for around 2 epochs. Then we compare the performance of our trained PPO model, denoted as PPO-story and the BC-old model on our evaluation set for short story generation tasks.

### 4.1.2 EVALUATION

Since we can not evaluate the responses of creative writing tasks automatically. We still use human annotators to compare the responses of the two models. In fact, our evaluation process is the same as the process we annotate our training data, except that prompts are changed to our evaluate sets.

To better eval the performance of our data, we evaluate the models on both in-domain and out-of-domain sets. For the in-domain sets, we generate 26 prompts using our prompt generation method. For the out-of-domain sets, we collect 19 human generated prompts which is quite different from our generated data.

We use two metrics to compare the performances of models. First we check the objective performances of models, i.e. the correctness and language scores. Since both scores range from 1 to 5. We get the ratio of responses that get both scores larger than 4. If one response gets 4 or 5 on both scores, we call this repsonse usable. This metric is denoted as usability. We also involve the subject judgement to compare the response. We directly compare the responses for one prompt and check the winner rate of the two models.

## 4.2 RESULTS

The usibilities of BC-old and PPO-story are shown in Table 2. Although the number of prompts are relatively small, we can still see a large improvement after the RLHF training. There is a improvement larger than 30 points on average. For out-of-domain prompts, the improvement is even larger than that on the in-domian prompts. We think this is because the out-of-domain prompts are relative easy and have fewer instructions than that in our generated prompts, making the improvement more obvious. We can see that RLHF training can improve the objective writing ability of our model obviously.

| Prompts type | BC-old | PPO-story |
|---|---|---|
| In-domain prompts | 19.23% | 53.85% |
| Out-of-domain prompts | 47.37% | 84.21% |
| All | 31.11% | 66.67% |

Table 2: The usabilities of models on both in-domain and out-of-domain prompts.

The winning rate between models, to some degree, represents metric combining subjective and objective judgement. We denote "G" as the annotator prefers response from PPO-story than that from BC-old. Otherwise, we denote the result as "B". Notice that we always ask annotators to distinguish two responses, and thus we do not have two responses preferred equally. The results are given below in Table 3. The result shows that PPO-story model outperforms BC-old model significantly. On in-domain prompts, PPO-story model is preferred slightly more than that on the out-of-domain prompts.

| Prompts type | G rate | B rate |
|---|---|---|
| In-domain prompts | 80.77% | 19.23% |
| Out-of-domain prompts | 76.32% | 23.68% |
| All | 78.89% | 21.11% |

Table 3: The winning rate of PPO-story over BC-old on both in-domain and out-of-domain prompts.

Both metrics above show that the creative writing training data through our EOPS method cause a big improvement after RLHF training.

### 4.3 THREE TASKS EXPERIMENTS

We further apply our EOPS pipline on two more tasks: the expand writing task and the style transfer task.

#### 4.3.1 SETTING

We collect more data for this experiments on three tasks. We use 26872 pairs for the short story generation task, 6695 pairs for the expand writing task and 7071 pairs for the style transfer task. We combine all these data with some general task data to conduct RLHF training process.

For this experiment, we use a new version of Baichuan 4 model, denoted as BC-new. This model has a better performance on creative writing than BC-old. We train both RM and PPO from BC-new model. Both training processes last for one epoch. We denote the RM and PPO trained models as RM-new and PPO-new respectively. We mainly compair the performances of BC-new and PPO-new on the three tasks.

Futher, we use the same preference data to train RM and PPO from Qwen2-72B model (Yang et al., 2024a). We denote the trained RM and PPO models as Qwen-RM and Qwen-PPO respectively. As mentioned above, our preferrence data on creative writing are collectted from different Baichuan models. Thus, there exists a mismatch of response distributions between Baichuan and Qwen models. From this experiment, we aim to see whether data collected from Baichuan model can perform well on Qwen models.

#### 4.3.2 EVALUATION

The evaluation for this experiment is almost the same as that in last experiment. We use human annotators and use usability and winning rate as our evaluation metrics.

Specifically, we have 94 prompts for the short story generation task, 95 prompts for the expand writing task and 100 prompts for the style transfer task. We compare the responses from BC-new, PPO-new, Qwen2-72B and Qwen-PPO.

#### 4.3.3 RESULTS

The usibility results are shown in Table 4.

Comparing BC-new and PPO-new, there are significant improvements on all three tasks. Specifically for short story generation where there are most training data, the improvement is most obvious. It can also be see that BC-new outperforms BC-new on the short story generation task. RLHF training can still bring a large improvement. On average, RLHF training with our generated data improve BC-new for about 7 points.

However, there are much less improvement on Qwen2-72B models. Althrough the improvement on short story generation is obvious, Qwen-PPO has a bad performance on the style transfer task. The results show that preference data collected from other models is not guaranteed to bring improvements.

| Prompts type | BC-new | PPO-new | Qwen2-72B | Qwen-PPO |
|---|---|---|---|---|
| Short story generation | 70.21% | 79.79% | 60.64% | 71.21% |
| Expand writing | 71.58% | 76.84% | 73.68% | 73.68% |
| Style transfer | 30% | 38% | 37% | 32% |
| All | 56.75% | 64.36% | 56.75% | 58.48% |

Table 4: The usabilities of models on three tasks.

For the winning rate between models, we denote a prompt to be "BC-G" if the annotator prefers response from PPO-new than that from BC-new, otherwise "BC-B". Similarly, we denote a prompt to be "Qwen-G" if the annotator prefers response from Qwen-PPO than that from Qwen2-72B, otherwise "Qwen-B". The results are given below in Table 5.

The results are quite similar to that on the utility. PPO-new outperforms BC-new on all three tasks. On Qwen models, however, RLHF training improves the ability of short story generation while hurting the other two tasks.

Notice the the winning rate of short story generation between PPO-new and BC-new is much smaller than that between PPO-old and BC-old. The reason may be that BC-new has much better performance and the improvement is harder for a better model.

| Prompts type | BC-G | BC-B | Qwen-G | Qwen-B |
|---|---|---|---|---|
| Short story generation | 59.57% | 40.43% | 62.77% | 37.23% |
| Expand writing | 56.84% | 43.16% | 49.47% | 50.53% |
| Style transfer | 57% | 43% | 45% | 55% |
| All | 57.79% | 42.21% | 52.25% | 47.75% |

Table 5: The winning rate of models on three tasks.

## 5 CONCLUSION

Our method mainly concentrate on how to produce high-quality human annocated data. We choose three tasks from creative writing. We propse an EOPS method to involve an expert to reduce the annotation noise. By involving the expert during prompt generation, annotation standard buiding and preference data generation process, we ensure the quality of our data. We conduct RLHF experiments with our data. The result shows that our data improves Baichuan model significantly. However, if we transfer these data to train Qwen models, the improvement is not ensured.

### AUTHOR CONTRIBUTIONS

If you'd like to, you may include a section for author contributions as is done in many journals. This is optional and at the discretion of the authors.

### ACKNOWLEDGMENTS

Use unnumbered third level headings for the acknowledgments. All acknowledgments, including those to funding agencies, go at the end of the paper.

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
