# OpenReview forum: "Data-Driven Creativity: Amplifying Imagination in LLM Writing"
_ICLR.cc/2025/Conference — Submitted to ICLR 2025_

### Official Review · Reviewer_uAk9 · 2024-11-03

**Soundness:** 1
**Presentation:** 1
**Contribution:** 2
**Rating:** 3
**Confidence:** 4

**Summary:**

The paper presents a method for improving the data collection process for human preference judgments to train LLMs on creative writing tasks. The crux of the method is that (amateur) crowd-workers might not have the expertise for annotating data for challenging tasks such as creative writing. The authors propose to use the task-specific knowledge of domain experts to create very fine-grained criteria/instructions to be given to crowd-workers when performing their annotations (framed as the Expert-Objective-Personal-Subjective (EOPS) method).

The criteria involve (a) 'correctness' - the authors extract multiple high-level _intentions_ from the user prompts (via an LLM) and the crowd-workers are asked to evaluate whether each of these intentions is satisfied, (b) 'language score' - the crowd-workers are explicitly asked to consider the plot, characters, expressivity of writing, depth of theme, emotional aspects, imaginativeness and logical rigor of the writing. Crowd-workers are expected to score the responses on a scale of 1-5 on both these criteria before making their preference ranking.

Experiments show that training the model on annotations collected via the EOPS method outperforms a baseline model (see Strengths/Weaknesses/Questions for comments).

**Strengths:**

1. The fundamental premise of the work is sound. For challenging creative writing tasks, we should be drawing from the domain knowledge of experts and using this to inform large-scale annotation projects. Connecting expert knowledge to annotators by distilling more fine-grained instructions/criteria for evaluation is done well.

2. The annotation format obtained from experts is pretty detailed and well-suited to the creative writing task. This could also be connected to earlier works on the theory of the process of writing (see question 4 below)

**Weaknesses:**

1. The work is missing two ablations that would improve the strength of the authors' arguments. The claim is that the EOPS pipeline improves the judgments provided by (amateur) annotators.
    (a) On a small subset of annotations, you could report the correlation between (i) judgments from experts and (amateur) annotators with the EOPS pipeline, and (ii) judgments from experts and (amateur) annotators without the criteria from the EOPS method. If (i) is higher than (ii) then it shows the value in the intermediate criteria from EOPS.
    (b) The RLHF results in Tables 3 and 5 compare the initial model (BC-old/new) with the model after PPO based on annotations from the (amateur) annotators. To make the claims that the EOPS method helps, you also need to compare it with training an RLHF model on annotators collected in the standard method via binary preference judgments

2. The current draft version has multiple gaps that would make it challenging to reproduce the results. Among these are (a) background and recruiting procedure for experts and crowd-workers, including the method of reaching out, inclusion criteria, and payment assigned for their time, (b) the sensitivity of the method to different prompts in Section 3.4, (c) L.301-303, it is mentioned that there was continuous communication to solve difficulties encountered during annotation, what were these difficulties, how was the contact maintained, (d) instances where the language is imprecise in terms of the model training/data collection eg. L.301, '5 to 7 annotators participated', L.322 ('around 2 epochs'), (e) confidence intervals for the results in the tables and significance testing (L.411) since the sample size is limited to < 100 examples, (f) non-standard model names i.e. BC-old and BC-new, is this Baichuan 1 and 2?

**Questions:**

1. The related work section can be improved by extending the discussion to more recent works and by further connecting some logical gaps. Some specific suggestions below:
    * L.73 - Needs a citation for the point that data collection for reward modeling being the bottleneck of RLHF. Consider [1] and the related work covered in that work
    * L.75 - While black-box LLMs do not disclose the data collection practices, it would be informative to cover the practices of data collection for open-source alignment datasets [2,3,4], most relevant of which is recent work on creative writing with LLMs [5]
    * L.78 - Rather than say 'specific' tasks such as reasoning and math, it would be more helpful to the reader to explain what makes creative writing tasks different from these analytical tasks, a definition of objective correctness - which would even directly connect to your method. Recent work also directly demonstrates that AIF might not be well suited to creative writing tasks [5, 6]
    * L.86 - To differentiate your work from Wang et al. (2024) it would be better to talk about the aspects of your data annotation process than highlight just that your method is for RLHF data, since the contribution is the annotation and not the training method.
    * L.90 - Perhaps mention that while other alignment algorithms exist, RLHF remains the predominantly used method which explains the choice to focus on the same. Also DPO takes in the same format so there's no need to scope down your contribution to just RLHF.

2. A high-level comment on framing is that the work differentiates between 'experts' and 'annotators'. I assume the distinction is actually between domain experts and _amateur_ crowd-workers recruited via traditional platforms like Prolific or Amazon Mechanical Turk. Is this correct? It would be helpful to clarify since the 'annotator' can be anyone regardless of expertise.

3. The paper heavily relies on domain expertise, which is great for moving alignment methods forward but it would be good to document the background of these experts and how the authors recruited them for this study. Are they professional creative writers or language teachers etc.?

4. L.104-107 - In addition to relying on experts for the breakdown, it would be good to cite work on the theory of writing since this breakdown also neatly lines up with the Flower model for writing. [7,8,9]

5. L.111 - What does 'capable of mass production' mean in the context of prompt writing?

6. Citing the appropriate Baichuan paper(s) would be helpful to the original authors [10]

7. L.159-L.161 - While I buy the intuition that accuracy in-distribution doesn't translate to better writing, is there an experiment or a citation to back up this statement?

8. L.162-165 - Does the breakdown come from the experts or the authors fo the work ('... we break down...')? Are these the same folks?

9. L.164 - Given that the annotators are making independent judgments, I think you intend to say based on the expert _breakdown_ of the task, and not the expert _opinion_?

10. When assigning 5-point scores for language score (L.272-274), how do you instruct annotators to weigh the different desiderata for scoring? I liked the breakdown for correctness in L.263-265, was there a specific scheme for this aspect?

11. L.343, 356, 426 - By 'usibilities/usabilities', do you mean the fraction of examples that have a score >= 4 on both aspects as noted in L.335-336?

12. General suggestions on grammatical errors:
    * L.73: proved -> proven
    * L.76: many work -> many works
    * L.85: \citep -> \citet
    * L.120: 'Table ??' -> Table 1
    * L.147-149 and L.175-177 are identical copies of one another. Consider contextualizing them more given the section they are present in so as to differentiate them.
    * L.154 says 'objective criteria' then L.156 says 'indicators' - Using consistent terminology across the work to make it easier for a reader to follow your writing
    * L.194 - Remove the stray 'w'
    * L.308 - ... we follow *the* RLHF ...
    * L.329 - eval -> evaluate
    * L.432 - 'utility'?

13. (minor) Can you clarify the significance of the title? I don't think 'imagination' is used anywhere else in the annotation schemas/instructions/prompts, barring as one of the many criteria in the 'language score'. Given the relative importance to the paper, perhaps a different title would be more suitable such as 'Fine-grained instructions for llm data collection in creative writing'

[1] Lambert, Nathan, et al. "Rewardbench: Evaluating reward models for language modeling." arXiv preprint arXiv:2403.13787 (2024).
[2] Kirk, Hannah Rose, et al. "The PRISM Alignment Project: What Participatory, Representative and Individualised Human Feedback Reveals About the Subjective and Multicultural Alignment of Large Language Models." arXiv preprint arXiv:2404.16019 (2024).
[3] Köpf, Andreas, et al. "Openassistant conversations-democratizing large language model alignment." Advances in Neural Information Processing Systems 36 (2024).
[4] Ethayarajh, Kawin, Yejin Choi, and Swabha Swayamdipta. "Understanding Dataset Difficulty with $\mathcal {V} $-Usable Information." International Conference on Machine Learning. PMLR, 2022.
[5] Chakrabarty, Tuhin, Philippe Laban, and Chien-Sheng Wu. "Can AI writing be salvaged? Mitigating Idiosyncrasies and Improving Human-AI Alignment in the Writing Process through Edits." arXiv preprint arXiv:2409.14509 (2024).
[6] Tuhin Chakrabarty, Philippe Laban, Divyansh Agarwal, Smaranda Muresan, and Chien-Sheng Wu. 2024. Art or Artifice? Large Language Models and the False Promise of Creativity. In Proceedings of the 2024 CHI Conference on Human Factors in Computing Systems (CHI '24). Association for Computing Machinery, New York, NY, USA, Article 30, 1–34. https://doi.org/10.1145/3613904.3642731
[7] Hayes, John R., and Linda S. Flower. "Writing research and the writer." American psychologist 41.10 (1986): 1106.
[8] Gero, Katy, et al. "A design space for writing support tools using a cognitive process model of writing." Proceedings of the first workshop on intelligent and interactive writing assistants (In2Writing 2022). 2022.
[9] Creativity Support in the Age of Large Language Models: An Empirical Study Involving Professional Writers. T Chakrabarty, V Padmakumar, F Brahman, S Muresan. Proceedings of the 16th Conference on Creativity & Cognition, 132-155, 2024
[10] Yang, Aiyuan, et al. "Baichuan 2: Open large-scale language models." arXiv preprint arXiv:2309.10305 (2023).

---

### Official Review · Reviewer_fujZ · 2024-11-04

**Soundness:** 3
**Presentation:** 2
**Contribution:** 2
**Rating:** 3
**Confidence:** 4

**Summary:**

This paper addresses the challenge of obtaining high-quality human preference data for training Large Language Models (LLMs) in alignment with human preferences, especially for creative tasks where preferences are complex and subjective. The authors propose the Expert-Objective-Personal-Subjective (EOPS) method, a framework that integrates expert guidance to reduce noise in the annotation process. EOPS is applied to three creative writing tasks, with experts assisting in prompt generation, establishing annotation standards, and guiding preference data collection. Experimental results show that EOPS data significantly enhances the performance of Baichuan models through Reinforcement Learning from Human Feedback (RLHF). However, the transfer of this preference data to Qwen models yielded less consistent improvements, highlighting potential model-specific dependencies in using EOPS data effectively.

**Strengths:**

(1) The research question in this paper is interesting. The paper attempts to design an effective data collection method for the challenging and complex task of creative writing and validates its effectiveness on the Baichuan model through RLHF training.

(2) The study explores multiple creative writing tasks, namely Short Story Generation, Expanded Writing, and Style Transfer.

**Weaknesses:**

(1) After training Qwen-2’s PPO using preference data collected from the four Baichuan models, the performance worsens or remains ineffective in "Expand writing" and "Style transfer" tasks. The authors conclude, “The results show that preference data collected from other models is not guaranteed to bring improvement.” This conclusion seems overly simplistic; evaluations on multiple datasets would be necessary. Moreover, ideally, a well-constructed preference dataset should be model-agnostic. If the author’s conclusion holds, it suggests that the dataset lacks diversity and that it may be beneficial to collect data from models beyond Baichuan.

(2) It’s unclear how the training and test sets are divided, as I didn’t find a detailed description of the test set’s construction or size. Only in line 330 is there mention of the number of prompts used for in-domain and out-of-domain sets.

(3) In Table 1, the results for the in-domain set are worse than for the out-of-domain set in both the original model and the PPO-trained model, which is confusing. Typically, the in-domain set is expected to be similar distribution to the model’s training set and thus perform better than the out-of-domain set, but the opposite conclusion is reached here.

(4) The paper contains typos that need careful revision. For example, in line 120, "Table ??" (possibly should be Table 1), and in line 63, "In summery" (likely meant to be "In summary").

**Questions:**

(1) What are the samples in the evaluation dataset like? (ICLR allows appendix submissions, so it's recommended that the authors include some examples, prompts, etc., in the appendix for readers.)

(2) How many test samples are there for the three evaluation tasks?

---

### Official Review · Reviewer_eauy · 2024-11-04

**Soundness:** 1
**Presentation:** 1
**Contribution:** 1
**Rating:** 1
**Confidence:** 5

**Summary:**

This paper introduces a new method for collecting high-quality human feedback data to improve large language models' creative writing abilities. The authors propose a data collection process called EOPS (Expert-Objective-Personal-Subjective) that combines expert knowledge with human annotator preferences. The process involves several steps: experts break down creative writing tasks into categories, AI helps generate diverse writing prompts, multiple annotators evaluate AI-generated responses using expert-defined criteria, and annotators provide rankings that combine objective scores with personal judgment. The authors tested their method on three creative writing tasks: short story generation, expanding existing text, and style transfer. Their results showed that models trained with this data achieved significant improvements on Baichuan models, and these improvements worked for both similar and different types of prompts than those used in training. However, when they tried using the same data to train Qwen models, the results were mixed.The paper's main contribution is introducing a structured way to collect human feedback data for improving AI creative writing, with a focus on combining expert knowledge with individual preferences to get better quality data.

**Strengths:**

I appreciate authors attempt in solving a real problem in AI development. The authors recognize that while automated feedback works well for tasks like coding or math, creative writing requires human judgment. Rather than simply collecting random human opinions, they sim to develop a structured process that combines expert knowledge with individual preferences.

**Weaknesses:**

1) The paper is really poorly written. A lot of the sentences are incoherent hard to understand. References to Tables are ?? (Line 120). Some sentences make no sense "It can also be see that BC-new outperforms BC-new on the short story generation task."

2) I find it baffling how the Intro and Related work has no citations on Creative Writing and only on RLHF. A paper that doesn't engage with literature is a fundamentally weak paper. There have been way too many papers on LLM and Creativity, some of which also discusses in depth the issues on how to design alignment.
(i) Are Large Language Models Capable of Generating Human-Level Narratives? Tian et al 2024
(ii) Pron vs Prompt: Can Large Language Models already Challenge a World-Class Fiction Author at Creative Text Writing? Marco et al 2024
(iii) Art or artifice? large language models and the false promise of creativity Chakrabarty et al 2024

3) Correctness score and Language Score are extremely weak metric for RLHF. In Creative Text Generation both response can have overlap between prompts and responses but that doesnt make one more desirable than each other. this is not a factuality eval. Language score sounds pretty adhoc to me as well. No theory / no justification as to why ? I recommend reading (iii) to reframe Language score

4) Only 5/7 annotators are not enough for robust findings especially if they are non experts. whats their background ? what is the agreement?

5) Why BC Old in one task and BC new in another ?

6) What are the potential biases in your alignment pipeline ? The metrics used are problematic. The "usability" metric (counting responses that score 4+ on both correctness and language) is a crude binary measure that loses important nuance. A response scoring 5/5 and 5/5 is treated the same as one scoring 4/5 and 4/5. The authors should use the full range of scores and report means and distributions.  Alignment training for subjective tasks needs to be aware of how desirable any individual response is, regardless of its preference relationship.

In general this is a very hastily written paper. The quality is not at all suitable for ICLR

**Questions:**

NA

---

### Official Review · Reviewer_eMMC · 2024-11-07

**Soundness:** 1
**Presentation:** 2
**Contribution:** 2
**Rating:** 3
**Confidence:** 4

**Summary:**

The goal of this paper is to produce better RLHF feedback data for RL (PPO) training using expert human guidance in conjunction with non-expert annotations in the feedback generation process. They have a multi-step data generation process with a structured sampling procedure for prompts, expert defined scoring categories and heuristics for scoring generated responses using LLMs. They evaluate by running their method to generate new synthetic data for finetuning the LLMs and use a reward model to optimize for PPO and show that largely their finetuning improves performance over the base model.

**Strengths:**

The paper suggests an interesting direction for collecting RLHF data, but it needs more work in proving that it is necessary and sufficient to follow the method suggested in the paper.

**Weaknesses:**

The paper can be made stronger in the following areas
* If the goal of the paper is to propose a better method for RLAIF, there should be a comparison with an RLAIF baseline. The authors compare the RLAIF trained model to the base model which doesn't say anything about the goodness of their method. This is critical for soundness.

* The method is difficult to piece together. Specifically
  * There is no example output or a figure that shows the process
  * The titles are confusing. For e.g. for section 3.3, I'm not sure how the text in that section corresponds to the title "Response Generation". It seems like a method for acquiring human preferences by using experts and non-professionals.
  * Where is section 3.4.1 used as part of the overall process?
  * There is almost an entire page (lines 216-256) of a model generated output but was not helpful in understanding the paper.

* The method is very elaborate, but there is no substantiation of the choices made. For e.g.
  * why is the specific style of sampling elements in Table 1 any good. How is it better than sampling a content prompt from an LLM. Are the "examples" chosen independent of each other? If so, how can we know they will lead to a cohesive output?
  * In section 3.4.1 why are the extracted themes better suited to be themes than the other content in the prompt? This seems like a very arbitrary choice.
  * Checking for these themes in Section 3.4.2 leads to checking for superficial keyword-match style scoring. i.e. covering some topics leads to no guarantees about how those topics are covered and the cohesion in the output.


* The authors don't state what is being shown to the reward model for training. Does it see more than just the human ratings? What human ratings is it trained with?


There are many weaknesses with lots of open questions that need to be answered in the spirit of good scientific practices.

**Questions:**

* line 149 - what are "conditional limitations"?
* line 320 - Why is the reward model small?
* How much data was used to train the reward model?
* line 331 - Why evaluate on only 45 prompts, is the reward model trained using roughly the same amount of data as well? It seems like a small data set given all the optimizations that have gone into using non-expert annotators in conjunction with expert annotators
* Where were the expert annotators sourced from?

---

### Meta-Review · Area_Chair_BD94 · 2024-12-17

**Metareview:**

The authors present a methodology alternative to RLAIF, that combines human experts, AI models, and human annotators aiming primarily at creative writing tasks. The reviewers acknowledge the importance of the topic, but raise several concerns. These include general lack of clarity of the paper and difficulty to follow, challenges in reproducibility, and lack of meaningful and rigorous evaluations.

**Additional Comments On Reviewer Discussion:**

No discussions, as the authors did not respond to the reviewers.

---

### Decision · Program_Chairs · 2025-01-22

Reject